# Molecular detection of *Coxiella burnetii* in raw meat samples collected from different abattoirs in districts Kasur and Lahore of Punjab, Pakistan

**Shahpal Shujat**[1,2], **Wasim Shehzad**[1], **Aftab Ahmad Anjum**[3], **Julia A. Hertl**[2], **Muhammad Yasir Zahoor**[1]*, **Yrjö T. Gröhn**[2]

**1** Institute of Biochemistry and Biotechnology, University of Veterinary and Animal Sciences, Lahore, Punjab, Pakistan, **2** Department of Population Medicine and Diagnostic Sciences, College of Veterinary Medicine, Cornell University, Ithaca, New York, United States of America, **3** Institute of Microbiology, University of Veterinary and Animal Sciences, Lahore, Punjab, Pakistan

* yasir.zahoor@uvas.edu.pk

**Data Availability Statement:** All relevant data are within the manuscript and its Supporting information files.

## Abstract

*Coxiella burnetii* is the zoonotic pathogen that causes Q fever; it is widespread globally. Livestock animals are its main reservoir, and infected animals shed *C. burnetii* in their birth products, feces, vaginal mucus, urine, tissues, and food obtained from them, i.e., milk and meat. There were previously very few reports on the prevalence of *C. burnetii* in raw meat. This study aimed to determine the prevalence of *C.burnetii* and its molecular characterization in raw ruminant meat from the Kasur and Lahore districts in Punjab, Pakistan, as this has not been reported so far. In this study, 200 meat samples, 50 from each species of cattle, buffalo, goat, and sheep, were collected from the slaughterhouses in each district, Kasur and Lahore in 2021 and 2022. PCR was used for the detection of the *IS1111* element of *C. burnetii*. The data were recorded and univariate analysis was performed to determine the frequency of *C. burnetii* DNA in raw meat samples obtained from different ruminant species using the SAS 9.4 statistical package. Of the total of 200 raw meat samples, *C. burnetii* DNA was present in 40 (20%) of them, tested by PCR using the *IS1111* sequence. The prevalence of *C.burnetii* differed among the studied species of ruminants. When species were compared pairwise, the prevalence in cattle was statistically significantly lower than in sheep (P = 0.005). The sequence alignment based on origin implied that the strains are genetically diverse in different districts of Punjab, Pakistan. The findings demonstrated that the prevalence of *C. burnetii*, especially in raw meat samples, deserves more attention from the health care system and professionals from Punjab, Pakistan, i.e., abattoir workers and veterinarians.

## Introduction

*Coxiella burnetii* is a gram-negative coccobacillus related to the Gamma subdivision of the Proteobacteria [1, 2]. It is an obligate intracellular bacterium replicating in eukaryotic cells and occurs in two forms [3]. The bacterium's large-cell variant (LCV) resembles the intracellular

**Funding:** This study was funded by International Research Support Initiative Program, Higher Education Comission, Pakistan fellowship. Grant:I -8/HEC/HRD/2021/11529 awarded to student. The funder has no role in study design, data collection and analysis, decision to publish, or preparation of manuscript.

**Competing interests:** The authors have declared that no competing interests exists.

replicative form. In contrast, the small-cell variant (SCV) is the non-replicating form of the bacterium that is released when the infected cells lyse and can resist long-term environmental stresses [1].

*Coxiella burnetii* is the zoonotic pathogen that causes Q fever worldwide [2]. It can infect different host species, including domestic, wild, and marine mammals, birds, reptiles, and arthropods. The main reservoirs of this infection are cattle, buffalo, goat and sheep, which are also a source of human infection [4]. It infects over 40 tick species, which is an important transmission vector in ruminants [5]. It replicates in ticks; thus sufficient amounts of this pathogen are eliminated in their faeces and deposited on the skin of animal hosts during feeding. Several studies have identified ticks as a potential risk for coxiellosis in livestock and other domesticanimals [6–10]. In humans, this infection can occur through inhalation of particles dispersed from environmental dust containing dried tick faeces [11–13] and direct contact with contagious wool, milk, meat, urine, semen and feces of animals [14]. In ruminants, infection can occur both asymptomatically and symptomatically. Clinical manifestation of this bacterium includes stillbirth, abortion, mastitis, endometritis, and other reproductive disorders in ruminants [15, 16]. Flu is a contagious respiratory illness caused by influenza viruses, fever, hepatitis, and endocarditis may also occur in humans [2, 17].

*Coxiella burnetii* is considered endemic and has a worldwide distribution, including Pakistan. It has gained international public health awareness with cases reported in humans throughout the globe, including 284 cases in Netherlands, 14 in Switzerland and 10 in France [16]. It has also been isolated from Australian abattoir workers [18]. Most cases remain undiagnosed due to a lack of proper diagnostic facilities in developing countries like Pakistan.

Molecular tests are commonly used to detect *C. burnetii* in samples of different origins, including blood, serum, milk, and meat. Several PCR assays have been used for the detection of *C. burnetii*. The *IS1111* gene is a frequently used PCR target and is considered more sensitive than single-copy gene targets for detection [19].

*Coxiella burnetii* is a neglected pathogen in Pakistan, although it substantially affects public health. Farm management and public awareness are required to control this infection. Moreover, the infection remains largely undetected, mainly due to limited diagnostic facilities, misdiagnosis for other diseases with similar symptoms (e.g., Brucellosis) and insufficient training of healthcare workers and clinical physicians in handling this contagious disease in developing countries like Pakistan. Notably, there have been only about six previous publications on human and animal Q fever from Pakistan in the international databank [20–25]. Information regarding *C. burnetii*'s manifestation in raw meat obtained from small and large ruminants for human consumption has not been collected so far. The objective of the current study was to estimate the prevalence of *C. burnetii* in raw meat samples collected from ruminants intended for human consumption in districts Kasur and Lahore.

## Methodology

### Study area and sampling

The Advanced Studies and Research Board at the University of Veterinary and Animal Sciences in Lahore, Pakistan approved this study in its 50$^{th}$ meeting held on 8 -02- 2019. The sampling was conducted between 2021 and 2022 and skeltal muscle meat samples were collected from the slaughterhouses of Districts Kasur and Lahore. This study included 200 skeltal muscle meat samples, 50 from each species of cattle, buffalo, goat, and sheep, from the slaughterhouses in each district, Kasur and Lahore.

## Sample processing and DNA isolation

Meat samples were stored at -20˚C and further processed for DNA extraction using a manual method [26]. 0.15 g of meat samples were minced and washed with distilled water and 70% ethanol and placed in a microcentrifuge tube. Then, 800 µl of Digestion buffer and 20 µl of Proteinase K, and 30 µl of 10% Sodium Dodiecyl Sulphate, remained pellet were added, and incubated at 56˚C overnight. After overnight incubation, 500 µl Phenol Chloroform Isoamylal-cohol was added and vortexed until the solution turned milky. Then it was centrifuged at 13500 RPM and 4˚C as mentioned above. Three layers were formed, and the upper transparent layer containing DNA was placed into a separate microcentrifuge tube. Two parts of isopropanol and 200 µl of chilled absolute ethanol were added in 1 part aqueous transparent layer and incubated for 20 min at -20˚C. It was then centrifuged under the same conditions, the supernatant was discarded, and the remained pellet. The taken pellet was washed using 200 µl of 70% ethanol and centrifuged under the same conditions. Then the supernatant was discarded, leaving the pellet for overnight drying to evaporate ethanol that act as a PCR inhibitor. The dried pellet was resuspended in 20 µl of distilled water in the water bath and heat shocked at 70˚C for 40 minutes. DNA quality and quantity were assessed using a spectrophotometer.

## Molecular assay and sequence analysis

*Coxiella burnetii* was detected using multiple copy gene amplification assay targeting the transposase gene, i.e., *IS1111*, and particular primers were used for this assay. The set of primers used for PCR amplification assay was sequenced as 5′–GTCTTAAGGTGGGCTGCGTG-3′ and 5`-CCCCGAATCTCATTGATCAGC-3` for forward and reverse primer [27]. The diagnostic assay was validated using Vircell Amplirun® Coxiella DNA Control. Each PCR reaction test contained 12.5 µl of 2X master mix, 1.25 µl of 10µM forward and reverse primer, and 1 µl of 50–100 ng DNA in a final volume of up to 25 µl by adding nuclease-free water. PCR reaction was performed using 96 well Applied Biosystems by Thermo Fisher Scientific thermal cycler. The reaction conditions were as follows: for initial denaturation at 95˚C for 5 min, denaturation at 94˚C for 30 sec, annealing at 60˚C for 30 sec, extension at 72˚C for 1 min, repeating steps 1 to 3 for 30 cycles, and final extension at 72˚C for 10 min. PCR products were analyzed on 2% agarose gel, and specific product was identified, i.e., 294 bp was observed during analysis. The obtained positive samples were sequenced commercially by Macrogen, Korea. Phylogenetic analysis was conducted on the sequences using the MEGA version 6.0 bioinformatics tool. Alignment and phylogenetic tree construction of 12 sequences, including two query sequences, were performed using the MEGA tool by the maximum likelihood method [28]. The nucleotide substitution model was verified and adjusted according to the data type and tamura nei model was selected for the analyzed data. Bootstrap value was adjusted as 100 number of replications.

## Data analysis

The data were recorded in a Microsoft Excel spreadsheet and univariate analysis was performed to determine the prevalence of *C. burnetii* in raw meat samples obtained from different ruminant sources using the SAS 9.4 statistical package. Chi-square tests were performed in PROC FREQ, and logistic regression models (PROC LOGISTIC) were fitted, with occurence of *C. burnetii* DNA as the outcome.

## Results

A total of 200 meat samples was assessed for *C. burnetii* DNA; it was found in 40 (20%) samples tested by PCR using the *IS1111* sequence. It was further distributed in four species of

**Table 1. Detection of *Coxiella burnetii* in meat samples collected from different species in Districts Lahore and Kasur, Pakistan, by PCR, 2021–2022.**

| Sample type | No. of examined samples | Number of positive samples | Percentage | P Value |
|---|---|---|---|---|
| Cattle | 50 | 6 | 12 | *Species* |
| Buffalo | 50 | 4 | 8 | 0.0008 |
| Goat | 50 | 11 | 22 | |
| Sheep | 50 | 19 | 38 | |
| Lahore | 100 | 20 | 20 | *Districts* |
| Kasur | 100 | 20 | 20 | 1.0000 |
| Total | 200 | 40 | 20 | |

ruminants as 8% in buffalo, 12% in cattle, 22% in goats, and 38% in sheep (Table 1). *Coxiella burnetii* DNA was thus found more frequently in the raw meat samples obtained from small ruminants (goats and sheep) compared to large ruminants (cattle and buffalo). *Coxiella burnetii* DNA was observed in 30% of mutton (sheep and goats) and 10% of beef (buffalo and cattle) samples. The *C. burnetii* prevalence differed significantly by species. There was no statistically significant difference between the two districts, however.

Chi-square tests showed that animal species, meat type, and animal age were associated with the occurence of *C. burnetii* DNA. Furthermore, logistic regression showed that meat samples from sheep were 7 times more likely to test positive than were samples from buffalo, 4.5 times more likely to test positive than were those from cattle and 2.2 times were more likely to test positive than from goat samples. When the 4 animal species were grouped into meat type (beef (buffalo, cattle) and mutton (sheep, goats)), mutton was 3.9 times more likely to test positive than was beef. Meat from 6-month-old animals was 3.9 times more likely to test positive than was meat from 1-year-old animals, and was 3.6 times more likely to test positive than was meat from 2-year-old animals (Table 2).

When the sequences were assessed on the basis of their origin, the results showed that in the current study, the sequences obtained from district Lahore were clustered with previously reported sequences from district Sahiwal because of their close genetic similarity. In comparison, the sequences obtained from district Kasur were clustered separately from those obtained from district Attock (Fig 1). Thus, the results showed that the strains in different districts of Punjab, Pakistan are genetically diverse (Fig 2).

## Discussion

The current study was designed to estimate the prevalence of *C. burnetii* in raw meat intended for human consumption. This study was the first to determine whether *C. burnetii* DNA occurs in raw meat from livestock animals. It included beef and mutton samples from cattle, buffalo, goats, and sheep from districts Lahore and Kasur, Pakistan. In previously reported

**Table 2. Detection of *Coxiella burnetii* in meat samples collected from different age groups in Districts Lahore and Kasur.**

| Animal Age | Prob | SE | Asymp.LCL | Asymp.UCL |
|---|---|---|---|---|
| 0.5y | 0.364 | 0.0649 | 0.2481 | 0.497 |
| 1 y | 0.129 | 0.0348 | 0.0748 | 0.214 |
| 1.5y | 0.174 | 0.0790 | 0.0668 | 0.382 |
| 2y+ | 0.333 | 0.2722 | 0.0434 | 0.846 |

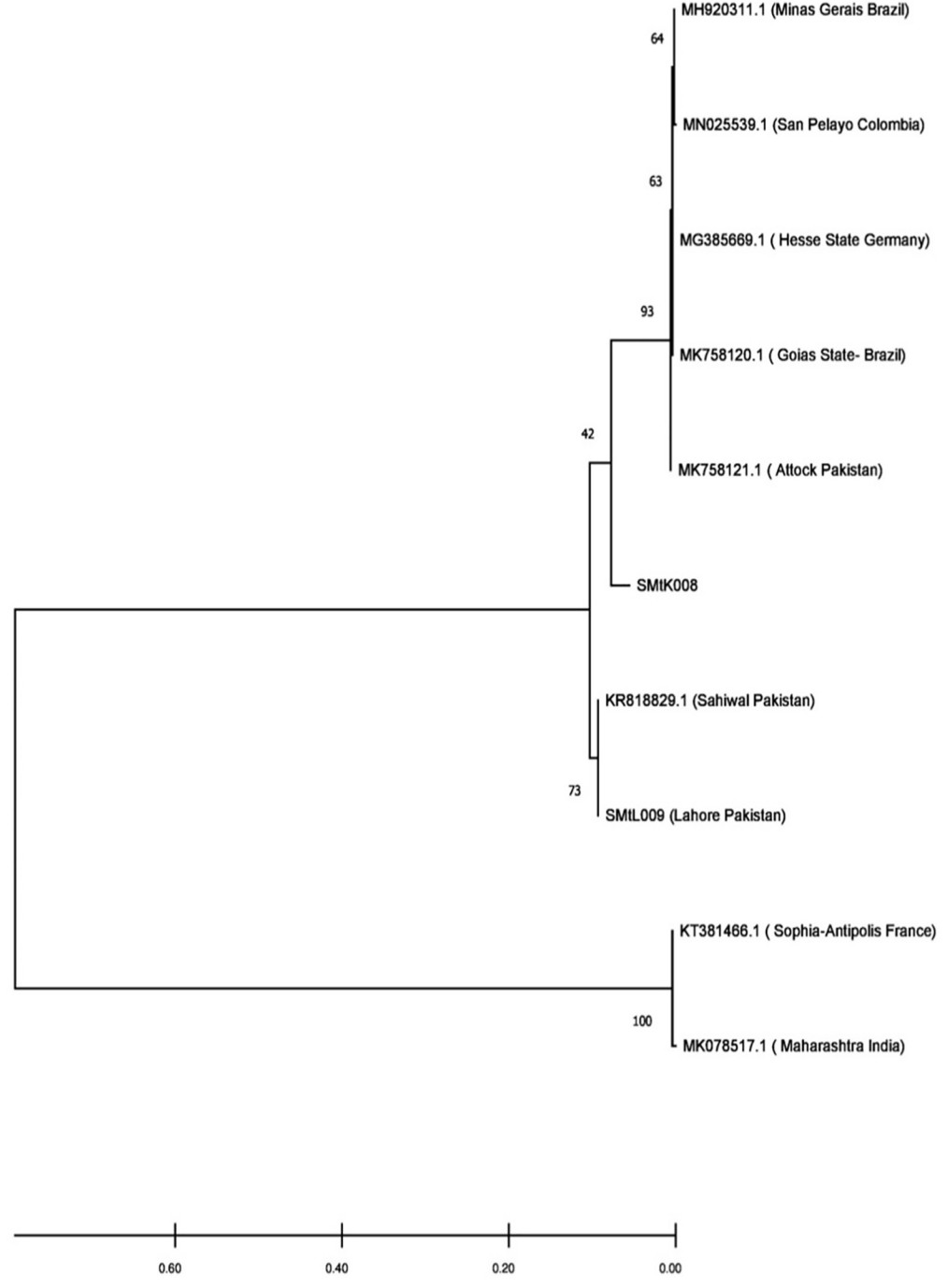

**Fig 1. The phylogenetic relationship of *C. burnetii* IS1111 gene sequence recovered from meat samples of district Lahore and Kasur, Pakistan, 2021–2022.** Labelled sequences are query sequences.

studies *C. burnetii* DNA was detected in food obtained from livestock animals, including milk and its products [29, 30]. The past studies suggested the risks for *Coxiella burnetii* through consumption of unpasteurized milk and its products are not negligible [31]. There were previously no reports of *C. burnetii* DNA occurrence in raw meat for human consumption. Past

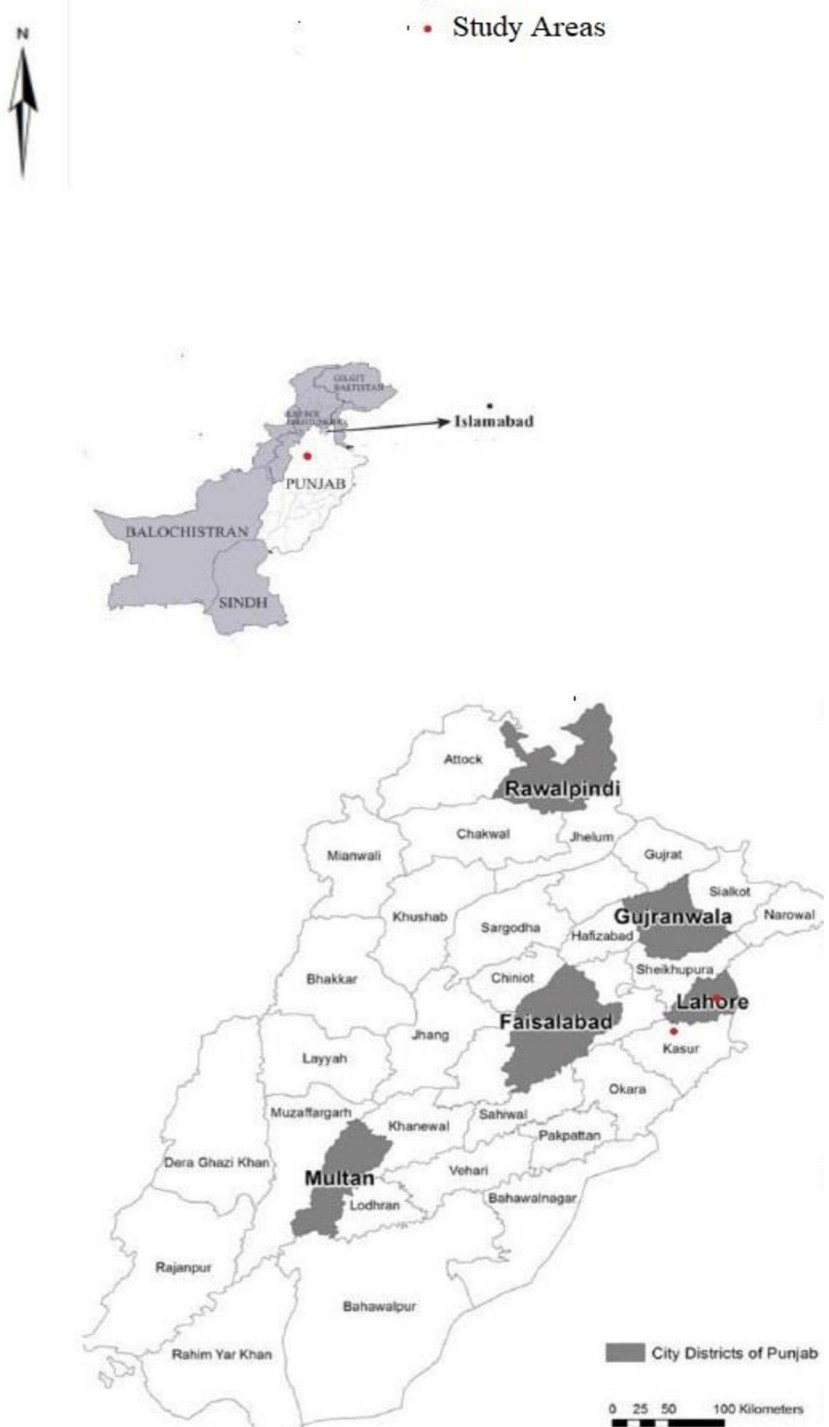

**Fig 2. The map showing all districts of Punjab, Pakistan.**

studies' limitations encouraged the current experiment to detect *C. burnetii* DNA in raw meat of livestock origin for human consumption.

A One Health approach was adopted in the current study to investigate raw meat samples from small and large ruminants. This study determined the prevalence of *C. burnetii* DNA to be 20% in raw meat samples collected from small ruminants. A previous study estimated the prevalence of *C. burnetii* in the blood samples of ruminants from district Kasur; it was estimated as 36.8% in small ruminants for the district Kasur [32]. For district Lahore, *C. burnetii* was reported in 4.8% of environmental samples [24]. *Coxiella burnetii* thus occurred more frequently in the blood of small ruminants than in raw meat.

In the current study, the estimated prevalence of *C. burnetii* in raw meat samples collected from sheep was 38%, while in districts Kasur and Lahore it was estimated to be 36% and 40%, respectively. In another study, the prevalence of *C. burnetii* in sheep blood was reported to be 46.9% in the Kasur district [23]. The prevalence of *C. burnetii* in sheep carcasses was reported to be 6.7% in Iran, while in other countries, there were no reports on sheep carcasses [24]. According to the findings, it is inferred that *C. burnetii* is common in sheep. Like goat meat, there is a substantial interest in consuming sheep meat and its products in Pakistan and other parts of the world. Therefore, paying attention to the food-borne pathogens in such communities is essential, and veterinary organizations must prioritize control and prevention strategies in livestock. The healthcare system should also provide training for at-risk people.

The current study found *C. burnetii* DNA in 22% of raw meat samples collected from goats. The prevalence of *C. burnetii* in districts Lahore and Kasur was 20% and 24%, respectively. There were no previous studies available for estimating *C. burnetii* DNA in meat samples obtained from goats. In contrast, *C. burnetii* prevalence was reported previously in cattle and sheep meat from Iran [33]. Goat meat is consumed in many countries. Therefore, serious attention must be paid to *C. burnetii* in goat meat.

*Coxiella burnetii* DNA was detected in 12% of cattle meat samples in this study. Its prevalence was estimated as 16% and 8% in districts Lahore and Kasur, respectively. In comparison, *C. burnetii* was present in 8% of overall buffaloes. Its prevalence for district Kasur was estimated as 8% and for district Lahore it was estimated as 8%. Previously, *C. burnetii* prevalence was reported in Iran in 5.7% of the samples collected from cattle carcasses [24]. Therefore, meat samples can be considered a source of *C. burnetii* in livestock animals. The limitations to the current study includes viability count and the remaining districts of the Punjab, Pakistan. In future studies, considerations must include the other districts of Punjab and risk evaluation in the human population for the infection, especially in professionals, i.e., abattoir workers and veterinarians. Based on our results, there was molecular evidence of *C. burnetii* in meat samples collected from livestock animals of the districts of Kasur and Lahore. These findings imply that *C. burnetii* prevalence, especially in meat samples, could pose a severe risk of Q fever to abattoir workers and consumers in Punjab, Pakistan.

## Conclusion

Molecular evidence of *C. burnetii* was observed in meat samples of cattle, buffalo, goats, and sheep collected from the slaughterhouses in two districts of Punjab, Pakistan. These findings emphasized that the prevalence of *C. burnetii*, especially in raw meat samples, deserves more attention from the health care system and meat industry in Kasur and Lahore of Punjab, Pakistan. Future studies must include other districts and risk evaluation in the human

population for the infection, especially in professionals, i.e., abattoir workers and veterinarians.

## Supporting information

**S1 Table. Univariable analysis of *Coxiella burnetii* in raw meat samples collected from slaughter houses in districts Kasur and Lahore in Punjab province, Pakistan, 2021–2022.** (DOCX)

## Acknowledgments

The authors are grateful to HEC, Assistant Director Livestock District Kasur Dr. Musarat, Veterinary assistant in Livestock Department, and Dr. Saqib, Veterinary Assistant, for assisting in sample collection.

## Author Contributions

**Data curation:** Shahpal Shujat, Wasim Shehzad, Aftab Ahmad Anjum.

**Formal analysis:** Shahpal Shujat, Julia A. Hertl, Yrjö T. Gröhn.

**Investigation:** Shahpal Shujat.

**Methodology:** Shahpal Shujat.

**Software:** Shahpal Shujat.

**Supervision:** Muhammad Yasir Zahoor.

**Writing – original draft:** Shahpal Shujat.

**Writing – review & editing:** Yrjö T. Gröhn.

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
