## [Decision Letter · Decision Letter 0]

13 Mar 2023

PONE-D-23-03478Molecular detection of Coxiella burnetii in raw meat samples collected from different abattoirs in districts Kasur and Lahore of Punjab, PakistanPLOS ONE

Dear Dr. Zahoor,

Thank you for submitting your manuscript to PLOS ONE. After careful consideration, we feel that it has merit but does not fully meet PLOS ONE’s publication criteria as it currently stands. Therefore, we invite you to submit a revised version of the manuscript that addresses the points raised during the review process.

We look forward to receiving your revised manuscript.

Kind regards,

Gizat Almaw

Academic Editor

PLOS ONE

Journal Requirements:

a) The name of the colleague or the details of the professional service that edited your manuscript.

b) A copy of your manuscript showing your changes by either highlighting them or using track changes (uploaded as a *supporting information* file).

c) A clean copy of the edited manuscript (uploaded as the new *manuscript* file).

"The authors received no specific funding for this work."

Reviewers' comments:

Reviewer's Responses to Questions

**Comments to the Author**

1. Is the manuscript technically sound, and do the data support the conclusions?

Reviewer #1: Partly

Reviewer #2: Partly

2. Has the statistical analysis been performed appropriately and rigorously? 

Reviewer #1: I Don't Know

Reviewer #2: Yes

3. Have the authors made all data underlying the findings in their manuscript fully available?

Reviewer #1: Yes

Reviewer #2: Yes

4. Is the manuscript presented in an intelligible fashion and written in standard English?

Reviewer #1: Yes

Reviewer #2: Yes

5. Review Comments to the Author

Reviewer #1: Overall, the present article is pertinent since it explored the frequency of C. burnetii and its molecular characterization in raw meat from ruminant sources in Pakistan. However, some items need attention and adjustment:

-Please, all paper needs a complete revision of references.

For example, references 1 and 4 are de same (duplicated)

Reference 26 is incomplete.

Line 39: About following affirmation and references: "It infects over 40 tick species, which is an important transmission vector in ruminants (5). It replicates in ticks; thus sufficient 41 amounts of this pathogen are eliminated in their faeces and deposited on the skin of animal hosts during feeding" It is recommended to review and address this issue more assertively and support it with recent references. The role of ticks as vectors of this bacterium is still controversial...

Line 71. The following phrase, "PCR was used to detect the IS1111 element of the C. burnetii genome" could be deleted from here.

Line 73: Is there any reference supporting the DNA extraction protocol from the meat sample? Please include it.

Line 87: To explain the PCR condition, please include the following:

-The final concentrations of the primers

-Reference of the type of taq used

- How were possible PCR amplification inhibitors in the sample evaluated?

Line 98 For the Phylogenetic analysis, please clarify with more detail. It needs to be improved:

-Was the Nucleotide substitution model (model test) verified and adjusted for the data before the ML method?

-Did Bootstrap? Please present it.

-Outgroup? Please include and present it.

Line 118: It is unclear in the methodology; how was it possible to verify the age of the animals from which the meat sample came?

In the discussion, the authors should address the issue of whether or not good meat cooking could inactivate this bacterium. Furthermore, how much could molecular detection be expected to ensure viable transmission of this species?

Reviewer #2: The manuscript entitled “Molecular detection of Coxiella burnetii in raw meat samples collected from different abattoirs in districts Kasur and Lahore of Punjab, Pakistan” investigates 200 meat specimens collected in slaughter houses in two districts of Punjab, Pakistan. Samples obtained from 4 species (goat, sheep,cattle and buffalo) were tested for the presence of specific C.burnetii sequence by PCR, also phylogenetic analysis was performed. The manuscript is easy to follow and provides valuable information on presence of C.burnetii DNA in raw meat samples – a subject that has been poorly known so far.

Major concerns:

1. The data about tested meat samples are too general. There is no information whether skeletal muscles or offal samples were collected. It might be an important issue, taking into consideration that one of the studies showed significant difference between prevalence of pathogen DNA in skeletal muscle meat and offal. Moreover there is lack of information if the same type of meat was collected from all animal species.

2. The data about age of tested animals are not present in the manuscript, but they are included as one of variables in chi square analysis.

3. Lack of essential information in phylogenetic tree i.e. scale bar and bootstap values of the nodes makes the figure non-informative. Once completed, the data should be analysed and discussed.

4. The discussion is a little bit one-sided, possible limitation of the research are not adressed. Moreover, the conclusion that „C. burnetii prevalence, especially in meat samples, pose a severe risk of Q fever” seems to be far-reaching. The presence of C. burnetii DNA was confirmed but the viability of bacteria in specimens was not evaluated and the risk level is unknown, so far. In adition, it would be advisable to refer to findings about milk and dairy products and importance of transmission via alimentary route.

Minor comments:

line 17: I suggest “in raw ruminant meat”

line 22: Please replace “ruminant sources” with “ruminant species”

line 26: I suggest “strains are genetically diverse (…)” rather than “strains are not genetically identical”

line 38: reservoirs of

line 46: Flu is a contagious respiratory illness caused by influenza viruses. Please correct

49: Delete dot after word France

line 52: Please correct as follows: to detect Coxiella burnetii in samples

lines 62-63: Please precise the aim of this study (add regions where prevalance was estimated)

line 63: Please correct “used for human consumption” to “intended for human consumption”

Methodology

It would be interested for readers to have more information about tested meat. Could you add information about type of tested meat? Were it skeletal muscle meat samples or offal? Did you collect the same type of meat from all animals? Is it possible to present some details i.e. types of offal or meat cuts that were incuded in the study?

line 68: between 2021 and 2022/in 2021 and 2022- please correct

line 69: I suggest: This study included 200 meat samples(…)

line 73: Please delete word temperature, it is unnecessary

line 75 and 76: Please add full names of the reagents SDS and PCI

line 77: Then it was centrifuged under the same conditions as mentioned above – Please complete the information about centrifugation conditions as none were mentioned in this section.

line 81: I suggest “remained pellet“

line 83: “(…) evaporate ethanol and act as a PCR inhibitor”- it is uncelar, please rephrase

line 83: I suggest: “pellet was resuspended”

Section molecular assay and sequence analysis

Please add information about controls that were used in DNA extraction and PCR.

line 87: Assay (not assays), please correct

lines 93-94: I suggest rephrasing the sentence e.g. The reaction conditions were as follows (…)

line 94: delete final

line 97: by Macrogen

line 100: Please add information about bootstap value.

lines 115-116: What about sheep vs. goat meat?

lines 118-120: The results of chi square analysis on animals age as one of variables were described, but the data about age of tested animals are not present in the manuscript. Please complete this information!

line 124: the sequences were clustered because their genetic similarity not geographical proximity, please rephrase to clarify.

lines 122-126: A map showing all district of the Punjab province would make easier for the readers to follow the analysis presented in this paragraph.

line 126: I suggest: genetically diverse.

Table 1: Please redesign the table and add information about age of animals (see comment for lines 118-120)

Figure 1: Phylogenetic tree has not got a scale bar and bootstap values of the nodes, please add them. Moreover, the tree would be more informative if you add the data about host orgin of the sequences and year of isolation.

line 142: I suggest: for human consumption

line 147: - the sentence is misleading. It was mentioned that in other study the prevalence of C.burnetii DNA was determined for blood samples collected from ruminants in Lahore and Kasur. Nextly you referred to publication that describes prevalence only in Kasur. The publication 19 ( also do not analyse the prevalence of C.burnetii DNA in animals from Lahore. Please correct the sentence or add appropriate publication.

line 152: Please correct the reference number 23 as the publication describes first isolation of pathogen from cheese in Brasil…

lines 154-156: It would be worthwhile to familiarize readers with culinary habits in Pakistan. What is the most consumed meat type in Pakistan? Do Pakistanis often consume raw meat and thus what might be the zoonotic risk?

lines 165-166: What was the prevalence for buffalo samples in each district (Kasur and Lahore)?

line 167: Please correct the reference number

References need to be adjusted to the requirements of the journal.

6. PLOS authors have the option to publish the peer review history of their article (what does this mean?). If published, this will include your full peer review and any attached files.

Reviewer #1: No

Reviewer #2: No

---

## [Author Response · Author response to Decision Letter 0]

1 Jun 2023

Thanks for the detailed review of the manuscript. All the suggestions are incorporated. The followings are the answer to the comments:

Reviewer 1

1. Overall, the present article is pertinent since it explored the frequency of C. burnetii and its molecular characterization in raw meat from ruminant sources in Pakistan. However, some items need attention and adjustment:

-Please, all paper needs a complete revision of references.

For example, references 1 and 4 are de same (duplicated) 

Reference 26 is incomplete. 

 Ans. The article has been reviewed and all the references are checked and adjusted accordingly. 

2. Line 39: About following affirmation and references: "It infects over 40 tick species, which is an important transmission vector in ruminants (5). It replicates in ticks; thus sufficient 41 amounts of this pathogen are eliminated in their faeces and deposited on the skin of animal hosts during feeding" It is recommended to review and address this issue more assertively and support it with recent references. The role of ticks as vectors of this bacterium is still controversial...

 Ans. Reviewed and provided with recent references.

3. Line 71. The following phrase, "PCR was used to detect the IS1111 element of the C. burnetii genome" could be deleted from here. 

 Ans. The phrase is deleted.

4. Line 73: Is there any reference supporting the DNA extraction protocol from the meat sample? Please include it. 

 Ans. Reference is incorporated.

5. Line 87: To explain the PCR condition, please include the following:

A. The final concentrations of the primers 

Ans. The working concentrations of primers used were 10 µM. It is incorporated in the manuscript. 

B. Reference of the type of taq used 

Ans. The taq was not used separately; it was already added in the master mix of Macrogen Company. 

6. How were possible PCR amplification inhibitors in the sample evaluated?

Ans. The samples were analyzed for determining contamination using UV spectrophotometer and concentration of DNA was also evaluated and DNA was diluted in working concentration before PCR.

7. Line 98 for the Phylogenetic analysis, please clarify with more detail. It needs to be improved :

A. Was the Nucleotide substitution model (model test) verified and adjusted for the data before the ML method? Data type, data alignment base tamura was suitable

Ans. Nucleotide substitution model was verified and adjusted according to the data set and Tamura Nei model was suitable for the analyzed data.

B. Did Bootstrap? Please present it. 

Ans. Bootstrap value was adjusted as 100 numbers of replications.

C. Outgroup? Please include and present it. 

Ans . The outgroup from obtained results were from the sequences obtained from France and India.

8. Line 118: It is unclear in the methodology; how was it possible to verify the age of the animals from which the meat sample came? (Changed to Line 126)

Ans. It was verified by adult teeth count method

9. In the discussion, the authors should address the issue of whether or not good meat cooking could inactivate this bacterium. Furthermore, how much could molecular detection be expected to ensure viable transmission of this species?

Ans. The data regarding inactivating pathogen while cooking meat has not been reported so far. The PCR method is fast, sensitive and specific means to detect disease causing pathogen. However, it cannot distinguish viable bacterial cells from dead cells or from free DNA or RNA in sample. In order to detect viable cells specific assays are developed that measure the production of species specific pre-RNA in samples. But the current study was designed on bacterial diagnosis not viable transmission. In this research we focused on the transmission of pathogen in butchers, veterinarians and meat handlers in kitchen rather than cooked meat.

Reviewer 2

Major concerns

1. The data about tested meat samples are too general. There is no information whether skeletal muscles or offal samples were collected. It might be an important issue, taking into consideration that one of the studies showed significant difference between prevalence of pathogen DNA in skeletal muscle meat and offal. Moreover there is lack of information if the same type of meat was collected from all animal species.

Ans. The skeletal muscle meat samples were collected for the study. The same type of meat was collected from the studied animal species. It is incorporated in the manuscript methodology section.

2. The data about age of tested animals are not present in the manuscript, but they are included as one of variables in chi square analysis. 

Ans. The data regarding age of the animals tested is incorporated in the manuscript.

3. Lack of essential information in phylogenetic tree i.e. scale bar and bootstap values of the nodes makes the figure non-informative. Once completed, the data should be analysed and discussed

Ans. The scale bar and the bootsrap values of the nodes are incorporated in the manuscript.

4. The discussion is a little bit one-sided, possible limitation of the research is not adressed. Moreover, the conclusion that „C. burnetii prevalence, especially in meat samples, pose a severe risk of Q fever” seems to be far-reaching. The presence of C. burnetii DNA was confirmed but the viability of bacteria in specimens was not evaluated and the risk level is unknown, so far. In adition, it would be advisable to refer to findings about milk and dairy products and importance of transmission via alimentary route.

Ans. The limitations of the study are incorporated in the discussion section. This study has no evidence in the previous records. Literature for meat samples does not show any viability rather calculated for future studies. The findings about milk and its products consumption risks are incorporated. The current study was focused on detection of pathogen but not distinguished between viable bacterial cells from dead cells.

Minorconcerns

1.line 17: I suggest “in raw ruminant meat”

Ans. Followed.

2. .line 22: Please replace “ruminant sources” with “ruminant species 

Ans. Replaced.

3. line 26: I suggest “strains are genetically diverse (…)” rather than “strains are not genetically identical” 

Ans.Followed.

4. line 38: reservoirs of 

Ans.Corrected

5. line 46: Flu is a contagious respiratory illness caused by influenza viruses. Please correct(Line 46 to 48)

Ans. Corrections are incorporated. 

6. 49: Delete dot after word France 

Ans.Deleted.

7. line 52: Please correct as follows: to detect Coxiella burnetii in samples 

Ans.Corrected

8. lines 62-63: Please precise the aim of this study (add regions where prevalance was estimated) 

Ans. The aim of the study is précised.

9. line 63: Please correct “used for human consumption” to “intended for human consumption”

Ans.Corrected

10.Methodology

It would be interested for readers to have more information about tested meat. Could you add information about type of tested meat? Were it skeletal muscle meat samples or offal? Did you collect the same type of meat from all animals? Is it possible to present some details i.e. types of offal or meat cuts that were included in the study?

Ans. The details regarding meat type are incorporated.

11. line 68: between 2021 and 2022/in 2021 and 2022- please correct 

Ans.Corrected.

12. line 69: I suggest: This study included 200 meat samples(…) 

Ans.Followed.

13. line 73: Please delete word temperature, it is unnecessary 

Ans.Deleted.

14. line 75 and 76: Please add full names of the reagents SDS and PCI 

Ans. Abbreviations are replaced with full names.

15. line 77: Then it was centrifuged under the same conditions as mentioned above – Please complete the information about centrifugation conditions as none were mentioned in this section.

Ans. The details are incorporated.

16. line 81: I suggest “remained pellet“ 

Ans.Followed.

17. line 83: “(…) evaporate ethanol and act as a PCR inhibitor”- it is uncelar, please rephrase

Ans. Rephrased.

18. line 83: I suggest: “pellet was resuspended”

Ans. Corrected.

19. Section molecular assay and sequence analysis

Please add information about controls that were used in DNA extraction and PCR.

line 87: Assay (not assays), please correct 

Ans. Corrected.DNA control details are mentioned in the manuscript.

20. lines 93-94: I suggest rephrasing the sentence e.g. The reaction conditions were as follows (…) 

Ans. Rephrased.

21. line 94: delete final

Ans. Deleted.

22. line 97: by Macrogen 

Ans. Corrected

23. line 100: Please add information about bootstap value. 

Ans. Added

24. lines 115-116: What about sheep vs. goat meat? 

Ans. Incorporated in the manuscript.

25. lines 118-120: The results of chi square analysis on animals age as one of variables were described, but the data about age of tested animals are not present in the manuscript. Please complete this information! 

Ans. The information regarding the age of the studied animals is incorporated in the manuscript.

26. line 124: the sequences were clustered because their genetic similarity not geographical proximity, please rephrase to clarify. 

Ans. Rephrased.

27. lines 122-126: A map showing all district of the Punjab province would make easier for the readers to follow the analysis presented in this paragraph. 

Ans. Incorporated.

28. line 126: I suggest: genetically diverse. 

Ans.Corrected

29. Table 1: Please redesign the table and add information about age of animals (see comment for lines 118-120) 

Ans. The table is designed for studied animal age.

30. Figure 1: Phylogenetic tree has not got a scale bar and bootstap values of the nodes, please add them. Moreover, the tree would be more informative if you add the data about host orgin of the sequences and year of isolation.

Ans. Scale bar and bootstrap values are incorporated.

31. line 142: I suggest: for human consumption 

Ans.Followed

32. line 147: -the sentence is misleading. It was mentioned that in other study the prevalence of C.burnetii DNA was determined for blood samples collected from ruminants in Lahore and Kasur. Nextly you referred to publication that describes prevalence only in Kasur. The publication 19 (also do not analyse the prevalence of C.burnetii DNA in animals from Lahore. Please correct the sentence or add appropriate publication.

Ans. Corrected.

33. line 152: Please correct the reference number 23 as the publication describes first isolation of pathogen from cheese in Brasil…

Ans. Added

34. lines 154-156: It would be worthwhile to familiarize readers with culinary habits in Pakistan. What is the most consumed meat type in Pakistan? Do Pakistanis often consume raw meat and thus what might be the zoonotic risk?

Ans. The skeletal muscle meat type is consumed more often and there is no report on raw meat consumption so it is not possible to explain the zoonotic risk.

35. lines 165-166: What was the prevalence for buffalo samples in each district (Kasur and Lahore)? 

Ans. Prevalence is added in the manuscript.

36. line 167: Please correct the reference number

References need to be adjusted to the requirements of the journal.

Ans. Corrected. I followed Vancouver style with little variation from the accepted paper.

---

## [Editor Report · Decision Letter 1]

23 Jun 2023

PONE-D-23-03478R1Molecular detection of Coxiella burnetii in raw meat samples collected from different abattoirs in districts Kasur and Lahore of Punjab, PakistanPLOS ONE

Dear Dr. Zahoor,

Thank you for submitting your manuscript to PLOS ONE. After careful consideration, we feel that it has merit but does not fully meet PLOS ONE’s publication criteria as it currently stands. Therefore, we invite you to submit a revised version of the manuscript that addresses the points raised during the review process.

We look forward to receiving your revised manuscript.

Kind regards,

Gizat Almaw

Academic Editor

PLOS ONE

Journal Requirements:

Additional Editor Comments:

The authors needs to strictly follow the 'Style and Format' of PLOS ONE. Their manuscript is a mix of Arial and Times New Roman. The authors need to revise and submit by strictly following the 'authors guideline'. eg. tables, figures.

---

## [Author Response · Author response to Decision Letter 1]

11 Jul 2023

Thanks for the review of the manuscript. The suggestions of academic editor and reviewers have been incorporated and article has been adjusted to the journal requirements accordingly i.e., Style and format are adjusted, and references are reviewed, and incomplete references are completed.

---

## [Editor Report · Decision Letter 2]

31 Jul 2023

Molecular detection of Coxiella burnetii in raw meat samples collected from different abattoirs in districts Kasur and Lahore of Punjab, Pakistan

PONE-D-23-03478R2

Dear Dr. Zahoor,

We’re pleased to inform you that your manuscript has been judged scientifically suitable for publication and will be formally accepted for publication once it meets all outstanding technical requirements.

Kind regards,

Gizat Almaw

Academic Editor

PLOS ONE